# Markerless mutagenesis enables isoleucine biosynthesis solely from threonine in *Methanothermobacter marburgensis*

Maximilian Klein,[1] Angus S. Hilts,[1,2] Ross T. Fennessy,[1] Nino Trattnig,[1] Thomas Stehrer-Polášek,[1] Simon K.-M. R. Rittmann,[1,2] Christian Fink[1]

**ABSTRACT** The archaeal model microorganism *Methanothermobacter marburgensis* has been studied for methane production for decades. However, genetic modifications are required to harness *M. marburgensis* for the generation of novel archaeal cell factories for industrial-scale production of commodity and high-value chemicals. Only the development of tools for genetic engineering opens up this possibility. Here, we present the establishment of the first markerless mutagenesis system for genetic modification of *M. marburgensis*. This system allows the recycling of positive selection markers and enables multiple sequential gene deletions or integrations. As a demonstration, we clarified the postulated isoleucine biosynthesis pathway directly from pyruvate via citramalate synthase (CimA). In doing so, we identified a putative CimA in *M. marburgensis* and deleted the CimA coding gene, resulting in auxotrophy for isoleucine. The complementation of *cimA* initiated through constitutive expression led to prototrophic growth similar to the wild type, demonstrating that *cimA* is essential for pyruvate-derived isoleucine biosynthesis in *M. marburgensis*. As it has been shown *vice versa* in *Escherichia coli* before, we were able to complement isoleucine biosynthesis with the integration of a synthetic isoleucine biosynthesis pathway from threonine for the first time in a methanogenic archaeon. This was achieved via genome integration of the characterized thermostable threonine deaminase from *Geobacillus stearothermophilus*. The successful integration of an alternative pathway for isoleucine production paves the road for future application of multi-gene biosynthetic pathways to overproduce industrially relevant chemicals.

**IMPORTANCE** The autotrophic, hydrogenotrophic methanogen *Methanothermobacter marburgensis* is one of the best-studied model organisms in the field of thermophilic archaea. The fact that *M. marburgensis* shows robust growth and scalability in bioreactor systems makes it a highly suitable candidate for industrial-scale bioprocesses. Additionally, the reported study provides the tools for genetic engineering that enable sequential genome modification in *M. marburgensis*. Scalable bioreactor cultivation, the ability to genetically engineer, and the recent discovery of natural amino acid secretion in *M. marburgensis* set the cornerstone for the generation of the first cell factories in archaeal biotechnology to economically produce carbon dioxide-derived commodity and high-value chemicals at industrial scale.

**KEYWORDS** archaea, methanogens, amino acid production, synthetic biology, biotechnology, metabolic engineering

Considering the current climate crisis, carbon dioxide ($CO_2$) emissions have to be drastically mitigated to limit the rise in global mean temperatures to less than 1.5°C, a goal set by the United Nations in 2021 (1). One solution to mitigate $CO_2$ emissions is through carbon capture and utilization systems, such as the direct use of $CO_2$ for carbonated drinks or fire extinguishers, or indirect usage for Power-to-Gas

Address correspondence to Christian Fink, christian@arkeon.bio, or Simon K.-M. R. Rittmann, simon.rittmann@univie.ac.at.

M.K., A.S.H., R.T.F., N.T., T.S.-P, S.K.-M.R.R, and C.F. are employees of Arkeon GmbH. S.K-M.R.R; is co-founder and shareholder of Arkeon GmbH.

See the funding table on p. 15.

(P2G) or Power-to-X (P2X) systems(2). P2G systems aim to store surplus electrical energy from renewable sources in the form of natural gas. The surplus of electrical energy is first converted to molecular hydrogen ($H_2$) and molecular oxygen via electrolysis. $H_2$ is then fed, together with $CO_2$, into a chemical (Sabatier reaction) or biological methanation process to yield pure natural gas (3, 4) that can be stored and distributed with the existing natural gas infrastructure. Biological P2X systems are based on the same principle ($H_2$ and $CO_2$ or syngas [$H_2$/$CO_2$/carbon monoxide as input]), but chemicals, rather than gas, are produced as a replacement for fossil fuels, precursors for cosmetics or pharmaceuticals, or human nutrition (2, 5). Up until now, biological P2X systems are mostly considered in the framework of gas fermentation by bacteria for industrial-scale production of ethanol or acetate.

While bacterial species have been considered for P2X approaches in a multitude of studies, archaeal species remain underrepresented in the field of biotechnological processes. Nonetheless, laboratory-scale proof-of-concept studies in archaeal biotechnology include geraniol production with *Methanococcus maripaludis*, polyhydroxybutyrate with *M. maripaludis*, 3-hydroxypropionic acid with *Pyrococcus furiosus*, and isoprenoid synthesis with *Methanosarcina acetivorans* (6–9). With respect to scaling up these processes, improvements were made in recent years for *M. maripaludis* in bioreactor systems (10). However, the best results for scale-up in gas fermentative bioreactor systems with methanogens were obtained using the highly robust thermophilic, autotrophic, and hydrogenotrophic methanogen *Methanothermobacter* spp. (11–13). One of the deepest studied species of *Methanothermobacter* spp. is *Methanothermobacter marburgensis* (14, 15). *M. marburgensis* grows optimally at 65°C, is highly resilient to pressure and shear forces, has a high specific growth rate (up to 0.69 $h^{-1}$), and a volumetric methane productivity of up to 945 mmol $L^{-1}$ $h^{-1}$. Taken together, these characteristics make it a suitable wild-type cell factory for bioprocesses (12, 16, 17). Recently, it has been shown that *M. marburgensis* naturally secretes amino acids in significant amounts, further evidence of its feasibility for industrial-scale P2X processes (18). A current prerequisite for the use of an organism as a microbial cell factory (i.e. amino acid production) is the ability to genetically engineer the microorganism to make them more commercially feasible. Although *M. marburgensis* meets many other requirements for the P2X process, up to now, genetic engineering has not been robustly demonstrated.

While methods for genetic modification of mesophilic methanogens had been developed by the late 1990s (19–21), genetic systems for thermophilic methanogens only progressed significantly in recent years. Basic genetic systems have been described for thermophilic methanogens, including *Methanocaldococcus* spp., *Methanoculleus thermophila*, and *Methanothermobacter thermautotrophicus* (22–24). While markerless mutagenesis systems for sequential genetic modification, such as deletion and complementation studies, are in place for *Methanocaldococcus* spp. and *M. thermophila,* it has not been demonstrated in *Methanothermobacter* spp. (23, 25). Despite the close relationship between *M. thermautotrophicus* and *M. marburgensis*, a robust protocol for genetic modification of *M. marburgensis* has not yet been developed. The only reported successful attempts for genetic modification were the generation of revertants of mutagenized amino acid auxotrophic strains via transduction of wild-type *M. marburgensis* genes with the virus ΨM2 (26). In addition to transduction, the natural uptake of high-molecular-weight genomic DNA from wild-type *M. marburgensis* has been reported to revert amino acid auxotrophy and generate 5-fluorouracil-resistant mutant strains (27, 28).

Besides the reversion of amino acid auxotrophic strains, the amino acid biosynthesis pathways of *M. marburgensis* have been deeply investigated. *M. marburgensis* is able to generate all 20 proteinogenic amino acids, making it a prototrophic organism (11). One amino acid of special interest, because it has two distinct biosynthesis pathways in bacteria and archaea, is isoleucine. Isoleucine is commonly synthesized from threonine via threonine deaminase (IlvA), which makes it a member of the aspartate family of

amino acids (29). However, in *Methanocaldococcus* spp. and several other species, it has been proven, on a genetic level, that isoleucine is synthesized from pyruvate by citramalate synthase (CimA), which is highly similar to isopropylmalate synthases for leucine biosynthesis (30, 31) (Fig. 1). In *M. marburgensis,* the pyruvate-derived isoleucine biosynthesis pathway was postulated based on the results of experiments using $^{13}$C-labeled pyruvate (32). However, the responsible enzymatic machinery is yet to be found and physiologically validated, as no *cimA* has been characterized in the genome.

To both generate the basic principles for a bioprocess of isoleucine production and to further develop our knowledge of *M. marburgensis* physiology, we established the first genetic system for *M. marburgensis,* including a system for markerless mutagenesis for sequential genetic modifications. With that, we were able to shed light on the isoleucine biosynthesis mechanism in *M. marburgensis* on a genetic level and to amend the microbe for an aspartate-derived isoleucine biosynthesis pathway through integration of a synthetic biosynthesis pathway in *M. marburgensis* via heterologous expression of *ilvA* from *Geobacillus stearothermophilus*. These latest developments continue to pave the way for the generation of future *M. marburgensis* cell factories in the framework of P2X processes.

## RESULTS

### Citramalate synthase in *M. marburgensis* identified through comparison to characterized CimA from *Methanocaldococcus jannaschii*

It has been shown that there are two known distinct ways isoleucine is biosynthesized in bacteria and archaea (29, 31). In the common biosynthesis pathway in bacteria, isoleucine is generated via threonine by IlvA (Fig. 1, dark orange). The alternative biosynthesis pathway of isoleucine known in archaea and few bacteria uses citramalate synthase, which converts pyruvate. This biosynthesis pathway has been confirmed in several methanogenic archaea including *M. marburgensis* (32–34). Conversion from pyruvate as a first reaction step is common among the branched-chain amino acids (isoleucine, leucine, and valine; Fig. 1, light blue). Both isoleucine biosynthesis pathways start with substrates derived from the central carbon metabolism and include allosterically inhibited enzymes for their respective products (Fig. 1, *cimA*, *ilvA*). Eikmanns et al. (32) proved that isoleucine is only pyruvate derived in *M. marburgensis* by $^{13}$C labeling of putative isoleucine substrates. This would suggest that there is an unknown citramalate synthase in the genome of *M. marburgensis*.

The initial search for archaeal *cimA* genes identified 2,067 candidate sequences across 1,002 genomes. After using the reciprocal best-hit approach, 731 sequences were retained (Fig. 2A). This research demonstrates that putative *cimA* genes are likely widespread across archaea, occurring across most of the haloarchaea, as well as in a large section of the Methanobacteriota, as previously shown (33). Results across the rest of the tree are more scattered yet are still found in nearly every major lineage. It should be noted that these sequences are unlikely to be exhaustive since the reciprocal best BLAST hit method can be sensitive to gene duplications and relatively stringent at times. Two sub-selections were made after the construction of the phylogenetic trees, based on similarity to the *M. marburgensis* and *M. jannaschii cimA* queries. The larger sub-selection included 338 of the sequences, while the smaller selection included 101 sequences. The selections, after re-alignment, formed a single sequence clade, grouping both *M. jannaschii* and a newly identified putative *cimA* from *M. marburgensis* (NC_014408.1_1099 in GTDB; 100% identical to WP_013295935.1; MTBMA_RS05455 locus in genome NC_014408.1).

The global alignment between the *M. jannaschii cimA* and the putative *cimA* from *M. marburgensis* showed an identity of 59.9% with a pairwise positive score of 78.0%, suggesting that they are homologous (based on the scoring matrix used; Fig. 2B). Two sequences are commonly considered homologous if they share 30% identity over their length, a cut-off that is supported by protein homology modeling (35, 36). In support of the suggested homology, a reciprocal best BLAST was used to compare the two

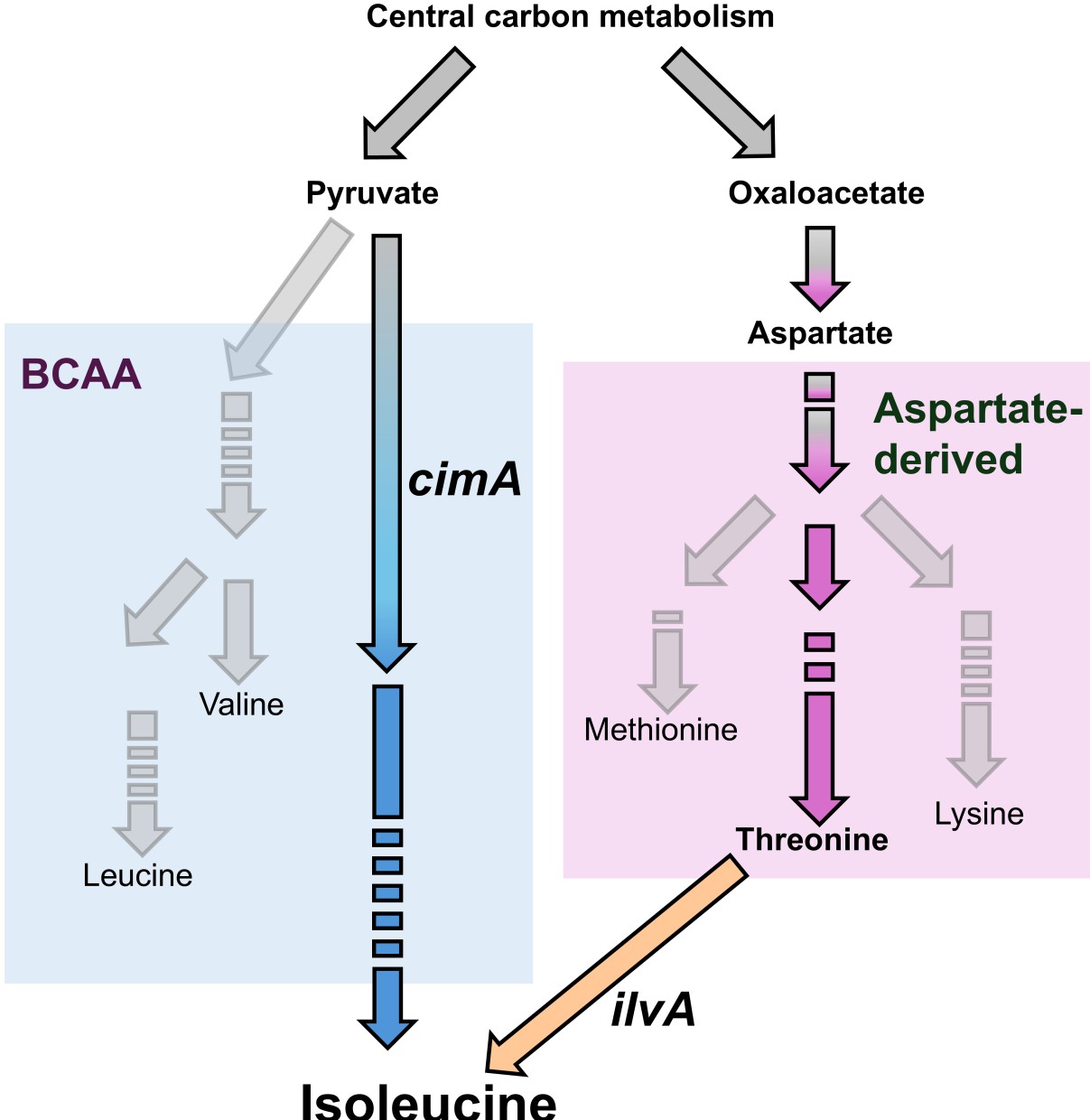

**FIG 1** Schematic of the biosynthesis pathways for amino acids of the aspartate family (purple) and branched-chain amino acids (BCAA in light blue), including substrate supply from the central carbon metabolism. The key enzymes for the two distinct isoleucine biosynthesis pathways are highlighted in dark orange (*ilvA*), for aspartate-derived biosynthesis, and in blue (*cimA*), for pyruvate-derived biosynthesis. Amino acids are marked as end-products.

genomes and identify if the two *cimA* sequences were one another's best matches. In both directions, this was the case (i.e., *cimA* from *M. jannaschii* most closely matched to the *M. marburgensis cimA* and *vice versa*).

## Deletion of the *hpt* gene enables markerless mutagenesis in *M. marburgensis*

To assess the product of the putative *cimA* gene in *M. marburgensis*, we needed to establish a genetic system for *M. marburgensis*. Hence, we set up a genetic system analogous to the system for *M. thermautotrophicus* based on conjugal DNA transfer from *Escherichia coli* S17-1 to *M. marburgensis* and genomic DNA integration through non-replicating vector constructs (37, 38). To this end, we designed the versatile vector system pArk, which is structurally similar to the commercially available pMTL80000

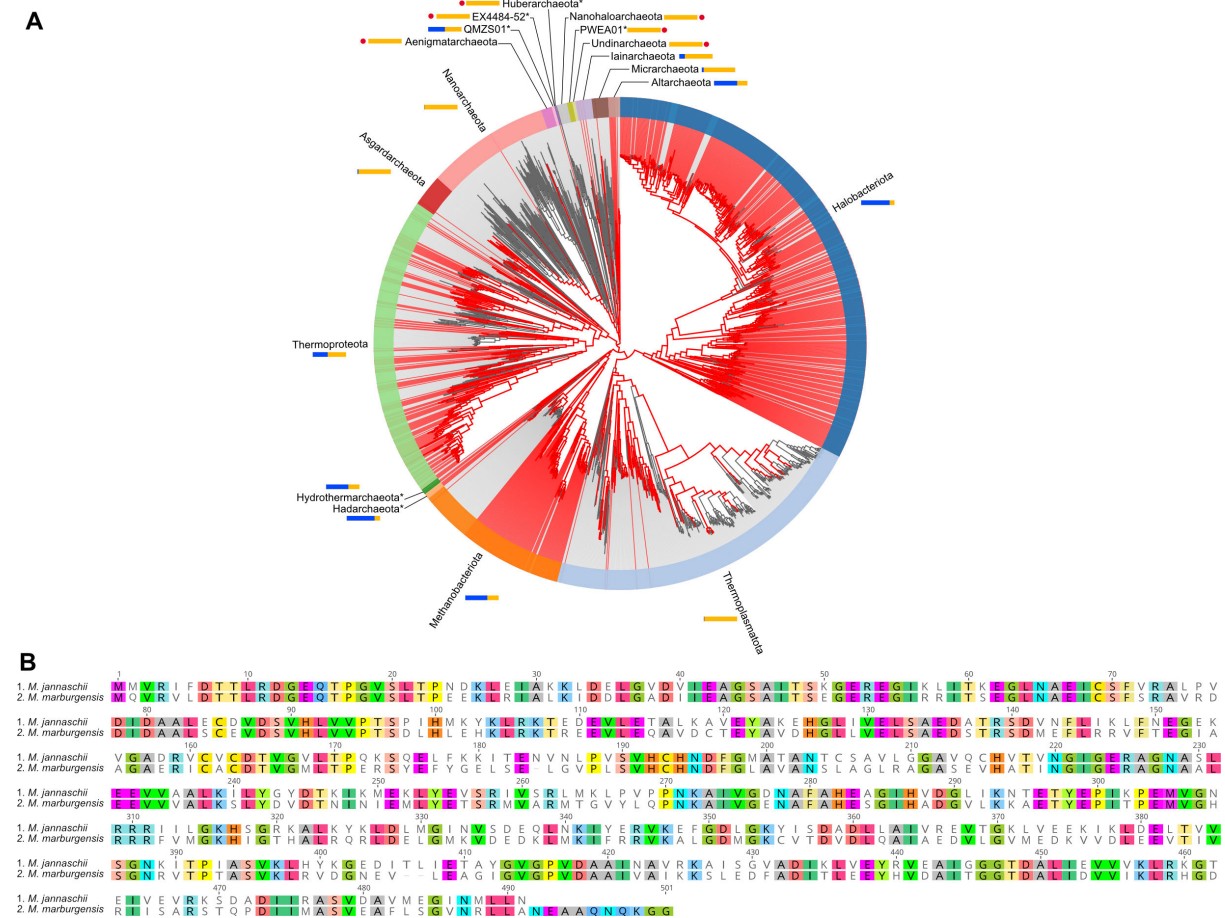

**FIG 2** Identification of a putative citramalate synthase in *M. marburgensis*. (A) Distribution of *cimA* genes across the archaeal tree of life, using an identity of 50% and the KOFAM model K09011. Red lines indicate a genome with the predicted genes, and colored bands around the tree indicate phyla. Blue bars represent the proportions of a phylum's members with the gene detected, and light orange bars represent the proportions for which it is missing. Red dots indicate phyla for which there were no detections, and asterisks (*) by the names indicate that the given phylum has less than 10 representative genomes. (B) An alignment of the putative *cimA* gene from *M. marburgensis* (NC_014408.1_1099; bottom) with the characterized *cimA* gene from *M. jannaschii* (NC_000909.1_1500; top). Where the sequence residues of both sequences match, they are colored according to which specific amino acid they are; otherwise, they are uncolored if they do not match.

Clostron system or pSEVA system based on unique restriction enzyme recognition sites (39, 40). In the pArk system, however, the restriction enzyme recognition sites were chosen based on the lowest number of restriction enzyme recognition sites in the genomic DNA of *M. marburgensis*. This makes the pArk system as versatile as possible in terms of the choice of homologous flanking regions for insertions and deletions (see Fig. S1A at [https://phaidra.univie.ac.at/detail/o:2120011]). Additionally, with its eight modules, pArk allows a high degree of flexibility for additional module integration in the future. The default vector pArk 2 is equipped with two modules for *E. coli*, positive selection via a chloramphenicol resistance gene, and a ColE1 high copy number origin of replication for *E. coli*, including the *traJ* region for efficient mobilization of the construct during conjugation (see Fig. S1A at [https://phaidra.univie.ac.at/detail/o:2120011]). The other six empty modules contain 100 bp of non-sense DNA fragments and can be replaced on demand with meaningful modules using restriction enzyme digest and ligation cloning approaches with the modular restriction enzyme recognition sites.

The first example for the exchange of modules is the integration vector for the hypoxanthine phosphoribosyltransferase (*hpt*) deletion to generate the base strain for future markerless mutagenesis mutants. As the exclusive purine recycling system in methanogens, Hpt has been proven to be an efficient system for markerless mutagenesis

for other species of methanogens, such as *M. maripaludis*, *Methanosarcina acetivorans*, and *Methanocaldococcus* sp. strain FS406-22 (25, 41, 42). A functional Hpt leads to the incorporation of 8-azahypoxanthine during nucleotide biosynthesis of methanogens when supplemented. This incorporation results in growth inhibition of the respective microbe. The deletion of *hpt*, however, results in resistance to 8-azahypoxanthine since the substance is no longer incorporated during nucleotide biosynthesis. Therefore, the reintroduction of the *hpt* gene and subsequent deletion thereof can be used as a powerful marker for negative selection (43). The two prerequisites to use *hpt* as a negative selection marker are the sensitivity of wild-type *M. marburgensis* to 8-azahypoxanthine and the existence of a Δ*hpt* strain of *M. marburgensis*. The sensitivity to 8-azahypoxanthine of *M. marburgensis* has been proven before (28). With that being known, the generation of the Δ*hpt* strain could be started. The annotated *hpt* (MTBMA_c17060) in the genome of *M. marburgensis* shares the same amino acid sequence as the characterized *hpt* gene from *M. thermautotrophicus* (44). Thus, we aimed for the deletion of MTBMA_c17060 for the generation of *M. marburgensis* Δ*hpt*. We generated a construct with 1 kbp homologous flanking regions upstream and downstream of *hpt* and left the start and stop codons, as well as two additional codons for amino acids, as a non-sense minigene. This should leave any potential operon structure intact. Additionally, we added a thermostable neomycin resistance with the constitutive promoter $P_{synth(BRE)}$ and a second copy of *hpt* with the constitutive promoter $P_{hmtB}$ (see Fig. S1B at [https://phaidra.univie.ac.at/detail/o:2120011]) (24). Although earlier studies report that deletion of *hpt* was achieved through a transformation with constructs only harboring homologous flanking regions of *hpt*, we were not successful with this strategy. Thus, we added a second constitutively expressed *hpt* to ensure sufficient expression and therefore complete inhibition of *M. marburgensis* growth when cultivated with 8-azahypoxanthine.

Markerless mutagenesis is a two-step process, where we first positively select, in our case via neomycin, for a single homologous recombination event of the construct with the genomic DNA. This results in a merodiploid state of genomic DNA where the entire *E. coli* backbone, the selective markers for *M. marburgensis*, the deleted state, and the native genomic state stay intact. We proved that the single homologous recombination event occurred with two primer combinations: (I) for an amplification of a part of the *E. coli* vector backbone and (II) to prove target-specific integration of the vector DNA at the *hpt* locus of *M. marburgensis* (Fig. 3A). With two homologous flanking regions, both upstream and downstream single homologous recombination is possible. In our case, we enriched an upstream single homologous recombined mutant. The subsequent negative selection with 8-azahypoxanthine resulted in *hpt* deletion mutants with residual wild-type signal in the PCR analysis (Fig. 3B). For further purification, we chose mutant 2 (Fig. 3B). When we performed a dilution series, wild-type PCR signals remained up to $10^{-5}$, but from $10^{-6}$, the signal was lost, indicating that only the desired *M. marburgensis* Δ*hpt* remained (Fig. 3C). Unspecific bands at sizes other than the expected band did appear, but neither the primer combination for inside the *E. coli* backbone nor the combination for the *E. coli* backbone to genomic DNA resulted in bands at the preliminarily identified sizes (Fig. 3C).

## Markerless deletion of *cimA* results in isoleucine auxotrophy

As a first proof of concept for the markerless mutagenesis method and to elucidate the isoleucine biosynthesis pathway in *M. marburgensis* on a genomic level, we chose the putative *cimA* (MTBMA_c11190) as a target for markerless deletion. CimA performs the first step in isoleucine biosynthesis from pyruvate. Hence, the deletion of *cimA* should result in isoleucine auxotrophy. To prove this hypothesis and exclude active alternative isoleucine biosynthesis pathways in *M. marburgensis*, we generated an integration construct based on the *hpt* deletion construct and exchanged the *hpt* homologous flanking regions with homologous flanking regions for *cimA*. In the first step after conjugation, we positively selected for single homologous recombination of construct

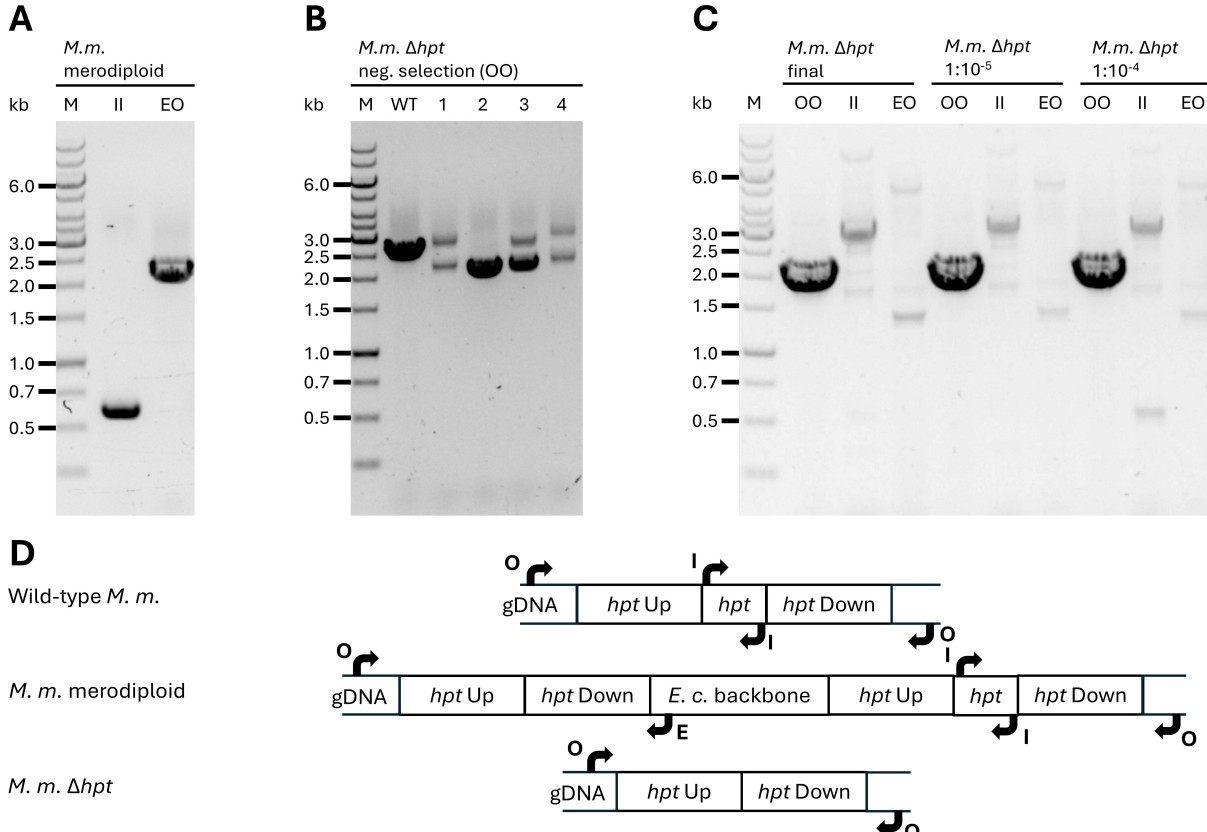

**FIG 3** PCR analysis of positive and negative selection for a pure *M. marburgensis* Δ*hpt* strain. (A) Depiction of the first merodiploid state of *M. marburgensis* with *E. coli* vector genome integrated with a single upstream homologous recombination event. II represents an inside-inside *hpt* vector primer combination for a PCR fragment. EO represents a primer combination specific for the *E. coli* vector backbone and native genomic DNA of *M. marburgensis*. (B) First analysis after negative selection with 8-azahypoxanthine with a primer combination outside of the *hpt* homologous flanking regions on genomic DNA of *M. marburgensis* (OO). While clones 1, 3, and 4 still show both the wild-type and deletion signals, the clone 2 signal points toward clean deletion of *hpt*. (C) PCR analysis of genomic DNA from *M. marburgensis* Δ*hpt* after a dilution series of 1:10,000 and 1:1,000,000 resulting in the final *M. marburgensis* Δ*hpt* pure strain. Abbreviations for primer combinations are the same across (A–C) and explained in panel (D). (D) Schematic drawing of genomic DNA states in wild-type *M. marburgensis*, *M. marburgensis* merodiploid, and *M. marburgensis* Δ*hpt*. Arrows and characters match the letters used in A–C and act as references for PCR fragment lengths.

pArk_00004 site-specific at the upstream or downstream flanking region of *cimA*, resulting in a merodiploid mutant of *M. marburgensis* Δ*hpt*. In the second step, we performed negative selection with 8-azahypoxanthine and simultaneously supplemented with 3 mmol L$^{-1}$ isoleucine to provide for potential isoleucine auxotrophy and generate the markerless *M. marburgensis* Δ*hpt* Δ*cimA* mutant (see Fig. S4A at [https://phaidra.univie.ac.at/detail/o:2120011]). After single colony enrichment of the *M. marburgensis* Δ*hpt* Δ*cimA* mutant, we made a dilution of 10$^{-6}$ to yield a pure culture. When we cultivated the *M. marburgensis* Δ*hpt* Δ*cimA* strain, we found no growth for up to 2 weeks of incubation at 62°C (Fig. 4A, solid blue line). When supplemented with 3 mmol L$^{-1}$ of isoleucine, however, the *M. marburgensis* Δ*hpt* Δ*cimA* strain showed wild type-like growth behavior (Fig. 4A, dotted blue line).

## Complementation of CimA restores the isoleucine biosynthesis pathway

As a next step, we wanted to gather further evidence for the function of CimA in *M. marburgensis*. Therefore, we complemented *M. marburgensis* Δ*hpt* Δ*cimA* with *cimA* expressed through the constitutive promoter P$_{glnA(M.v.)}$ from *Methanococcus vannielii*. This glutamine synthetase promoter (P$_{glnA(M.v.)}$) has been demonstrated to initiate high gene expression in *Methanococcus maripaludis* (45). In previous experiments, we also showed high expression levels of P$_{glnA(M.v.)}$ in *M. marburgensis* (data not shown). To

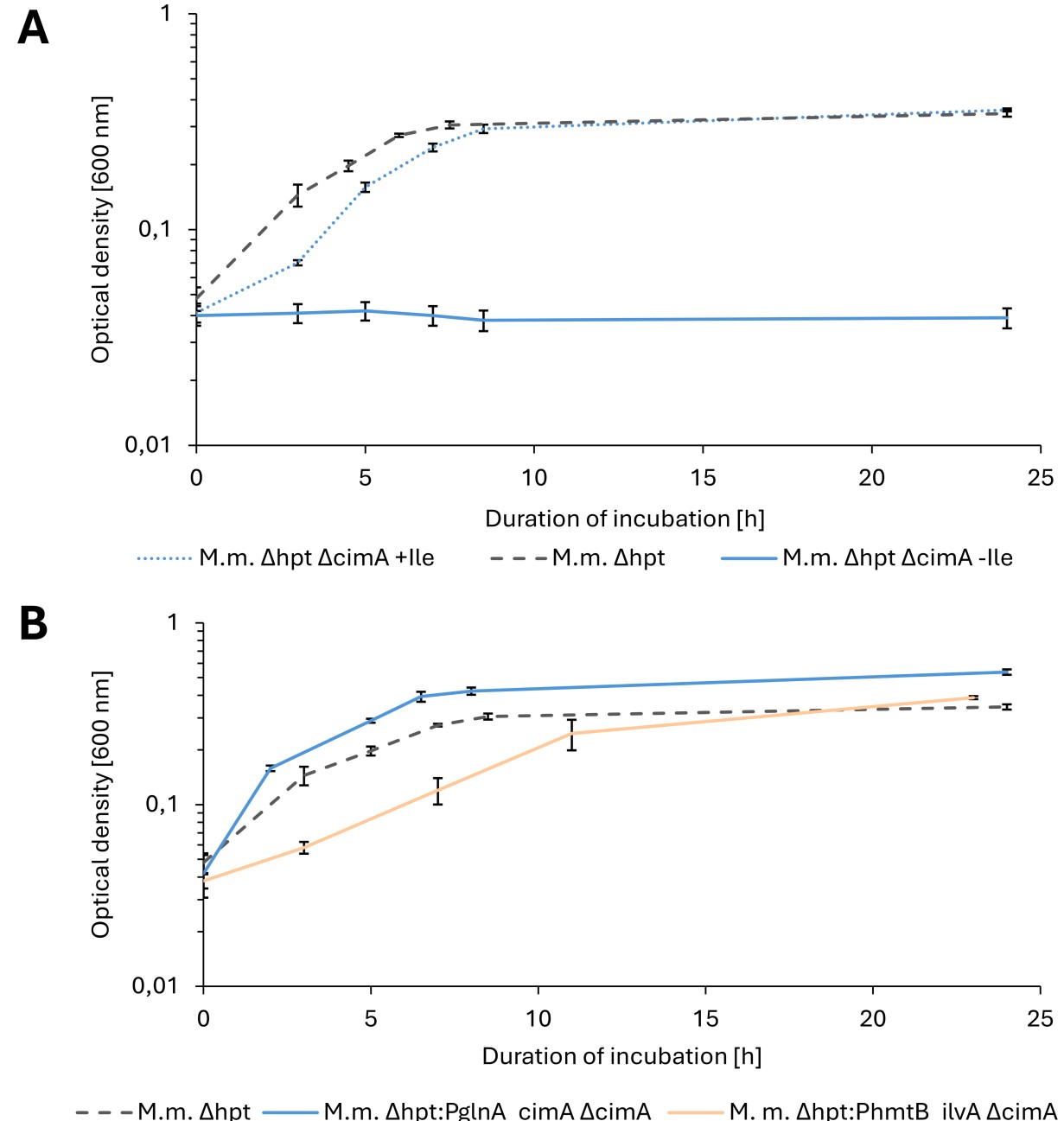

**FIG 4** Growth of *M. marburgensis* mutants (*M.m.*) n half-logarithmic depiction. (A) Comparison of *M. marburgensis* Δ*hpt* (dashed gray line) and *M. marburgensis* Δ*hpt* Δ*cimA* with (dotted blue line) and without 3 mmol L$^{-1}$ isoleucine (blue line) supplementation. (B) Comparison of wild-type *M. marburgensis* Δ*hpt* (dashed dark gray line) and *M. marburgensis* Δ*hpt*: P$_{glnA(M.v.)}$_*cimA* Δ*cimA* (blue line) without isoleucine supplementation as well as *M. marburgensis* Δ*hpt* (dashed-dotted light grey line) and *M. marburgensis* Δ*hpt*: P$_{hmtB}$_*ilvA* Δ*cimA* (light orange line) without isoleucine supplementation. Black bars indicate standard deviations (*n* = 4, unsupplemented deletion mutant *n* = 2). The *y*-axis represents optical density values at 600 nm, and the *x*-axis is the incubation duration at 60°C in h. This compiled graph comprises two phenotypic characterization experiments that were carried out on different days and with different batches of the same media; therefore, two *M. marburgensis* Δ*hpt* curves are depicted.

facilitate the subsequent PCR analysis after transformation, we decided to complement the *cimA* at the *hpt* deletion site. Therefore, we generated a construct with the *cimA* and the constitutive P$_{glnA(M.v.)}$ promoter in the opposite reading direction of the deleted *hpt* gene and placed the gene of interest between the *hpt* homologous flanking regions. Due to the markerless deletion in *M. marburgensis* Δ*hpt* Δ*cimA*, we were able to recycle our positive selection marker NeoR. Thus, we could again select for the citramalate

complementation mutant of *M. marburgensis* Δ*hpt* Δ*cimA* with neomycin. In this case, we only selected for the first step, the merodiploid *M. marburgensis* Δ*hpt*: P$_{glnA(M.v.)}$_*cimA* Δ*cimA* mutant, since the putative gain of function of isoleucine biosynthesis is not dependent on a negatively selected mutant. We purified the merodiploid strain with upstream integrated P$_{glnA(M.v.)}$_*cimA* and the *E. coli* backbone and did not find any residual *M. marburgensis* Δ*hpt* Δ*cimA* signals (see Fig. S4B at [https://phaidra.univie.ac.at/detail/o:2120011]). In a phenotypic characterization of growth behavior of *M. marburgensis* Δ*hpt*: P$_{glnA(M.v.)}$_*cimA* Δ*cimA* compared to *M. marburgensis* Δ*hpt*, we found that the prototrophy was restored and isoleucine biosynthesis was functional again. The growth behavior of *M. marburgensis* Δ*hpt*: P$_{glnA(M.v.)}$_*cimA* Δ*cimA* seemed slightly different compared to *M. marburgensis* Δ*hpt*, with a trend to faster growth during the exponential growth phase (Fig. 4B). To exclude the possibility of growth due to isoleucine carryover from isoleucine-supplemented precultures, we performed high-performance liquid chromatography-mass spectrometry (HPLC-MS) -based amino acid quantifications of the culture broth at the same time points as for optical density measurements. We measured isoleucine levels of 0.5 mg L$^{-1}$ at inoculation, and we consider this concentration as not sufficient for growth supplementation (see Fig. S6A at [https://phaidra.univie.ac.at/detail/o:2120011]). Additionally, the isoleucine concentration increases over time, which is additional evidence for a functional isoleucine biosynthesis pathway (see Fig. S6A at [https://phaidra.univie.ac.at/detail/o:2120011]).

### Complementation of isoleucine biosynthesis with thermostable threonine deaminase from *G. stearothermophilus* enables biosynthesis from threonine

A second possibility for complementation of the isoleucine biosynthesis pathway is the threonine-derived alternative, via IlvA (Fig. 1). To guarantee functionality under thermophilic conditions, we chose the characterized thermostable IlvA from *Geobacillus stearothermophilus* and codon-optimized the underlying *ilvA* gene for use in *M. marburgensis* (46, 47). As was done for the *cimA* complementation, we integrated the *ilvA* gene with a constitutive promoter (P$_{hmtB}$) into the deleted locus of *hpt*. We generated the *ilvA*-containing mutant using the isoleucine auxotrophic strain *M. marburgensis* Δ*hpt* Δ*cimA* (see Fig. S5A and B at [https://phaidra.univie.ac.at/detail/o:2120011]). When 3 mmol L$^{-1}$ of isoleucine was supplemented, the *M. marburgensis* Δ*hpt* Δ*cimA* strain grew wild type-like (data not shown). However, without isoleucine supplementation, there was no growth visible after several days of incubation. Therefore, we decided to supplement with 3 mmol L$^{-1}$ of threonine to supply substrate for isoleucine biosynthesis and restore growth. When we transferred this threonine-adapted *M. marburgensis* Δ*hpt*:P$_{hmtB}$_*ilvA* Δ*cimA* to non-supplemented media, prototrophic growth was restored and isoleucine biosynthesis was functional again (Fig. 4B). To exclude false-positive prototrophic growth behavior due to isoleucine carryover from isoleucine-supplemented precultures, we performed amino acid quantifications of the culture broth at three time points during incubation. We measured isoleucine and threonine levels close to the minimal detection limit in the beginning of the exponential growth phase, and we consider this concentration as not sufficient for growth supplementation (see Fig. S6B at [https://phaidra.univie.ac.at/detail/o:2120011]). Additionally, the isoleucine concentration increases over time, which is, as for *M. marburgensis* Δ*hpt*: P$_{glnA(M.v.)}$_*cimA* Δ*cimA,* also additional evidence for a functional isoleucine biosynthesis pathway via IlvA (see Fig. S6B at [https://phaidra.univie.ac.at/detail/o:2120011]).

### DISCUSSION

In this study, we describe the first genetic system for *Methanothermobacter marburgensis* for gene deletion and integration. We demonstrated markerless deletion of the *cimA* via negative selection through the reintroduction of the native *hpt*. This allows for the recycling of the positive selection marker to generate sequential genetic modifications, such as the implementation of synthetic metabolic pathways, gene complementation, or multiple gene deletions. As a first application of this system, we generated an

isoleucine auxotrophic mutant of *M. marburgensis* via deletion of *cimA*. Afterward, we restored isoleucine biosynthesis by complementation of the auxotrophy with *cimA* with the constitutive promoter P$_{glnA(M.v.)}$ and by constitutive heterologous expression of a thermostable threonine deaminase gene (*ilvA*) from *G. stearothermophilus*. With that, we proved the functionality of an alternative heterologous isoleucine biosynthesis pathway in *M. marburgensis*.

To enable markerless mutagenesis, the generation of an *M. marburgensis* Δ*hpt* mutant strain was necessary. The traditional approach to transform *M. marburgensis* with a construct that contains homologous flanks of *hpt* and to positively select with 8-azahypoxanthine was not successful for us (41). Therefore, we first integrated a constitutively expressed copy of *hpt* next to the native gene via neomycin positive selection. In a second recombination event, we removed both copies of *hpt*, the recombinant and the native one, with 8-azahypoxanthine negative selection. During the negative selection, we faced issues with the generation of a pure strain of *M. marburgensis* Δ*hpt*. For a more stringent negative selection, it was shown that 6-methyl-purine can be a highly efficient alternative to 8-azahypoxanthine that will be tested in future experiments (48, 49)

The fact that deletion of *cimA* in *M. marburgensis* resulted in auxotrophy for isoleucine proves that *M. marburgensis* synthesizes isoleucine via the less common citramalate synthase pathway, on a genetic level. This matches the results of a previous study, where evidence was gathered using $^{13}$C labeling, showing that isoleucine was synthesized from pyruvate (via CimA) in *M. marburgensis,* rather than threonine (via IlvA)*,* as is common in bacteria (32). CimA as part of the LeuA family of proteins evolved in three different clades (50): the *Leptospira* spp., the polyphyletic group of methanogens, and a third polyphyletic clade with members of archaea and bacteria, such as *Geobacillus* spp., *Pyrococcus* spp., and *Thermus thermophilus* (50). The presence of *cimA* is not directly connected to thermophily since it also occurs in mesophilic methanogens. It could, however, have been spread in the polyphyletic group via horizontal gene transfer, since most members of this clade share the trait of thermophily (50).

After proving that isoleucine is synthesized from pyruvate in *M. marburgensis*, we aimed to restore the biosynthesis pathway. This was to lend further evidence that isoleucine was derived from pyruvate as well as to explore the possibility of synthesis from threonine. Complementation with a constitutively expressed *cimA* resulted in wild type-like growth without supplementation of isoleucine (Fig. 4B). This means that prototrophic growth was regenerated, and there is no significant burden for *M. marburgensis* to express *cimA*. Additionally, the regeneration of prototrophic growth was possible with the expression of *ilvA*. The growth rate with *ilvA,* however, was slightly reduced compared to wild-type *M. marburgensis*. One reason could be the usage of a lower-strength constitutive promoter with *ilvA* compared to *cimA*. Another possibility could be due to energy constraints, as it was also reported that pyruvate-derived isoleucine production is less energy intensive than via threonine (51). Because of this, CimA is generally used in application-based generation of cell factories for isoleucine production and intermediates thereof (52, 53). Regardless, we could prove that both pathways are feasible to restore prototrophic growth in *M. marburgensis*. In *Geobacter sulfurreducens,* which harbours both metabolic pathways, it was proven that the carbon flux to isoleucine via pyruvate exceeded the flux via threonine. Additionally, *G. sulfurreducens* grew slower with *cimA* deleted, which was not the case when *ilvA* was deleted (50). This reflects our own results in *M. marburgensis* and could suggest that the *ilvA* pathway is evolutionarily older, but that clades which picked up the *cimA* gene were conferred an energetic benefit, therefore suppressing *ilvA*'s role.

Besides the advantages of using CimA in biotechnology, it can also enhance the robustness of systems for genetic engineering, since they rely on potent markers for positive selection. It has been reported that *Methanothermobacter* spp. are resistant to most commonly used antibiotics or that antibiotics are not stable at 65°C (54). Neomycin is one of the antibiotics that shows strong inhibition with up to 99.99% in *M. thermautotrophicus* (24). However, spontaneous resistance is a concern and has been observed

in both *M. thermautotrophicus* and *M. marburgensis* (24). One possibility to reduce the number of spontaneously resistant mutants is to rely on auxotrophic markers instead. Markers for amino acid biosynthesis are very common for this, since protein biosynthesis is inhibited by the absence of any individual amino acid. Therefore, the *cimA* deletion and complementation with *cimA* or *ilvA* will be highly potent markers for positive selection in *M. marburgensis*.

In conclusion, the proof of the functional CimA and IlvA pathways for isoleucine production, in addition to their use as a selection marker, further enhances the use of *M. marburgensis* for biotechnological purposes. Taking this together with the ability of *M. marburgensis* to secrete amino acids (18) and the possibility of overcoming allosteric inhibition as shown in heterologously expressed CimA of *M. jannaschii* in *E. coli*, overproduction of allosteric resistant CimA or IlvA could lead to archaeal cell factories for isoleucine production. Additionally, having two distinct pathways to choose from provides the freedom to potentially limit by-product formation and enhance substrate availability in archaeal cell factories for other aspartate family- or pyruvate-derived amino acids.

## MATERIALS AND METHODS

### Microbial strains

We used *Escherichia coli* NEB5α (New England Biolabs, Ipswich, MA, USA) for molecular cloning purposes and storage of constructs. We purchased *E. coli* S17-1 (DSM 9097) from *Deutsche Sammlung für Mikroorganismen und Zellkulturen* (DSMZ, Braunschweig, Germany) for conjugal transfer of DNA. We performed genetic engineering on *Methanothermobacter marburgensis* Arkeon (DSM 34858), the proprietary strain of Arkeon GmbH, which shows high genetic similarity to *M. marburgensis* Marburg (DSM 2133). All *M. marburgensis* strains used in this study are listed in Table 1.

### Cultivation conditions

We cultivated *E. coli* in premixed Luria-Bertani (LB) broth (Lennox) (Carl Roth GmbH + Co. KG, Karlsruhe, Germany) where necessary supplemented with 30 µg mL$^{-1}$ chloramphenicol for positive selection. Furthermore, we added 10 µg mL$^{-1}$ of trimethoprim to stabilize the *tra* gene cassette in *E. coli* S17-1. For solidified LB media plates, we added 15 g L$^{-1}$ of kobe agar (Carl Roth GmbH + Co. KG, Karlsruhe, Germany) to liquid premixed LB broth (Lennox) before autoclaving. We cultivated liquid cultures at 37°C in a shaker incubator with 150 rpm (Labwit, Burwood East (VIC), Australia) and cultures on solidified LB plates in a static incubator at 37°C (Binder GmbH, Tuttlingen, Germany).

We cultivated *M. marburgensis* Arkeon under anoxic conditions in minimal media. Therefore, we generated two variants of minimal media. On the one hand, we used sea salt (MS) media for transformation-related cultivation, such as precultures for transformation, outgrowth (recovery) after transformation, or selective enrichment. We prepared MS media as previously described in Fink et al. (24) with the following modifications: instead of a 0.2% (wt/vol) $(NH_4)_2Ni (SO_4)_2$ solution, we used 1 mL of a 0.02% (wt/vol) $NiCl_2$ solution. Furthermore, as a reducing agent and sulfur source, we used 0.2% (vol/vol) of a 0.5 mol L$^{-1}$ $Na_2S \cdot 9H_2O$ stock solution. For the variation of MS medium for conjugal spot mating, we added 2.5 g L$^{-1}$ of peptone and 1.25 g L$^{-1}$ of yeast extract during the media preparation, resulting in the complex media composition LB/MS. For phenotypic characterizations in closed batch and precultures thereof, we used *M. marburgensis* media, with the chemical composition as described before in Abdel Azim et al. (16) with the minor modification of adding 4 mL of a 0.025% (wt/vol) resazurin solution as a redox indicator.

We prepared both media compositions as described in Fink et al. (24). In brief, we generated 1 L of salt medium and set the pH to 7.2. As a first step of anaerobizing, we gassed with molecular nitrogen/$CO_2$ (20 Vol.-% $CO_2$ in molecular nitrogen) for 40 min, and afterward, we transferred to anaerobic medium to the anaerobic chamber (Coy

**TABLE 1** *M. marburgensis* strains and vector constructs used in this study

| Strains/Plasmids | Characteristics | Reference |
|---|---|---|
| *M. marburgensis* Arkeon | | |
| Wild-type | Wild-type isolate of *M. marburgensis* Arkeon (DSM 34858) | DSMZ 34858 |
| Δ*hpt* | Base strain of *M. marburgensis* for markerless mutagenesis with *hpt* negative selection | This study |
| Δ*hpt* Δ*cimA* | *CimA* deletion strain with auxotrophy to isoleucine | This study |
| Δ*hpt*:P$_{glnA}$_*cimA* Δ*cimA* | *CimA* complementation strain | This study |
| Δ*hpt*:P$_{hmtb}$_*ilvA* Δ*cimA* | IlvA complementation strain | This study |
| Plasmid constructs | | |
| pArk 2 | Base vector for genome integration in *M. marburgensis* | This study (55) |
| pArk00001 | Base vector with codon-optimized neomycin resistance | This study |
| pArk00002 | Base vector with homologous flanking regions for *hpt* deletion | This study |
| pArk00003 | *hpt* deletion vector with neomycin resistance for positive selection and *hpt* for negative selection | This study (55) |
| pArk00004 | *CimA* deletion vector with neomycin resistance for positive selection and *hpt* for negative selection | This study (55) |
| pArk00005 | Integration vector at the *hpt* deletion site for genes of interest | This study |
| pArk00006 | Integration vector at the *hpt* deletion site for *cimA* complementation with *cimA* | This study (55) |
| pArk00007 | Integration vector at the *hpt* deletion site for *cimA* complementation with *ilvA* | This study |

Laboratory Products, Grass Lake, MI, USA). The medium was reduced inside the anaerobic chamber and afterward filled anaerobically to 20 mL for MS medium and to 25 mL for *M. marburgensis* medium in 120 mL serum bottles. After sealing the serum bottles with butyl stoppers and aluminum crimps, we exchanged the headspace to $H_2/CO_2$ (4:1) (20 Vol.-% $CO_2$ in $H_2$) with three cycles of −0.8 bar and 1 bar overpressure with a final pressure of 1 bar $H_2/CO_2$ (4:1) gas mixture. The purity of all gases was 5.0 (99.999%). We autoclaved ready-made media bottles at 121°C for 20 min and 1.2 bar above atmospheric pressure in a vertical autoclave (HMC Europe GmbH, Tüssling, Germany).

To solidify media for *M. marburgensis*, we added 1.5% (wt/vol) of bacto-agar (Becton, Dickinson and Company, Franklin Lakes, NJ, USA) prior to autoclaving. We cultivated liquid cultures at 62°C in a shaker incubator with 170 rpm (Eppendorf New Brunswick Innova, Hamburg, Germany). Cultures on solidified minimal media were incubated in an anaerobic steel jar with 1 bar overpressure of $H_2/CO_2$ (4:1). We reduced the atmosphere inside the jar by adding 0.2 mL of a 0.5 mol L$^{-1}$ Na$_2$S·9H$_2$O solution on a paper towel. Anaerobic jars were routinely incubated in a static incubator at 62°C, except if stated otherwise.

## Primer, constructs, and synthesized fragments

All primers for this study were purchased from Eurofins Genomics (Eurofins Genomics Europe Shared Services GmbH, Ebersberg, Germany) (see Table S1 at [https://phaidra.uni-vie.ac.at/detail/o:2120011]). All constructs used in this study are shown in Table 1. The codon-optimized neomycin resistance cassette, including promoter and terminator, and the base vector pArk 2 were synthesized at Azenta Genewiz (GENEWIZ Germany GmbH, Leipzig, Germany), and the promoters P$_{hmtB}$ and P$_{glnA(M.v.)}$ gBlocks were ordered at IDT Genomics (Integrated DNA Technologies, Inc., Coralville, IA, USA) (see Table S2 at [https://phaidra.univie.ac.at/detail/o:2120011]).

## Molecular cloning methods in *E. coli*

We performed all PCR amplifications with Q5 high-fidelity polymerase (New England Biolabs, Ipswich (MA), USA) with suitable primer combinations (see Table S1 at [https://phaidra.univie.ac.at/detail/o:2120011]) according to the manufacturer's protocol. However, we reduced the primer concentration by four times and split all samples into 20 µL maximal volume for PCR and pooled them afterward again. All restriction enzymes were purchased from NEB (New England Biolabs, Ipswich, MA, USA) and used according to the manufacturer's protocol. We carried out DNA purification with the Qiagen PCR purification kit (Qiagen, Hilden, Germany). We performed DNA ligation with

the instant sticky-end ligase master mix (New England Biolabs, Ipswich, MA, USA). We halved the sample size and used a ratio of 1:6 for vector and insert. We transformed *E. coli* with chemically competent cells and a standard heat-shock protocol (56). We analyzed transformed *E. coli* cultures with a Qiagen Miniprep kit (Qiagen, Hilden, Germany) and restriction digests with specific restriction enzymes suggested by SnapGene software. We sent out constructs for final confirmation on a per-base level for Sanger Sequencing with Eurofins Genomics (Eurofins Genomics AT GmbH, Vienna, Austria).

## Construct design of *E. coli*

All constructs were generated via restriction digestion and ligation cloning with materials in Table 1 (see Table S1 at [https://phaidra.univie.ac.at/detail/o:2120011], and see Table S2 at [https://phaidra.univie.ac.at/detail/o:2120011]). The detailed cloning strategy is given in Supplemental materials S2 at https://phaidra.univie.ac.at/detail/o:2120011.

## Genetic modification of *M. marburgensis* Arkeon

We performed the genetic modification of *M. marburgensis* Arkeon according to the protocol for *Methanothermobacter thermautotrophicus* ΔH with slight modifications (38). In brief, 1 mL of an overnight culture of pArk construct carrying *E. coli* S17-1 is transferred to 10 mL of selective LB broth (Lennox) with trimethoprim in a 50 mL baffled flask. When this culture is grown to $OD_{600}$ of ~2.5, we harvest the culture at 3,000 $g$ for 15 min at room temperature. The supernatant is discarded via decanting, and the residual supernatant is removed with a pipet. The pellet is transferred into the anaerobic chamber (Coy Laboratory Products, Grass Lake, MI, USA) and gently mixed with 8 mL stepwise anaerobically centrifuged *M. marburgensis* overnight culture resuspended in 250 µL of anaerobic medium. A total of 100 µL of the cell mixture is spotted on an LB/MS solidified media plate. After the spot is fully absorbed, the plate is transferred into an anaerobic steel jar, gas exchanged to 1 bar overpressure $H_2/CO_2$ (4:1) and incubated for 16 h at 37°C. After the incubation time, the dried spot is washed off in 1 mL of sterile MS media and transferred to 4 mL of sterile non-selective media in a 25 mL serum bottle, followed by 3–4 h of incubation at 62°C shaking at 170 rpm (Eppendorf New Brunswick Innova, Hamburg, Germany). Then, 1 mL of the recovered *M. marburgensis* cell suspension is transferred to 20 mL of 250 µg mL$^{-1}$ neomycin sulfate-containing selective MS media. The culture is incubated rotating at 170 rpm at 62°C in a shaker incubator. If growth occurs before 60 h of incubation, the genetic modification is rendered successful.

## Generation of pure mutant strains

To purify a PCR-confirmed *M. marburgensis* mutant strain from wild-type or merodiploid strain residues, the first selective liquid enrichment was transferred to 20 mL of sterile selective minimal media. This second transfer was plated onto selective solidified MS media plates to obtain single colonies. Three to five single colonies were picked with a needle and directly transferred into selective liquid MS media. The resulting single colony liquid enrichments were serially diluted to a final dilution of 1:2 million in selective MS media. Genomic DNA was prepared from 3 mL of the highest diluted enrichment and analyzed via PCR to check for residual wild-type background. If no wild-type signals were observed, compared to a control, the strain was considered pure. Additionally, we tested all negatively selected pure strains for no growth on neomycin for at least 8 days.

## Markerless mutagenesis of *M. marburgensis* Arkeon

We used pArk00003 for the first markerless mutagenesis approach. The homologous flanking regions are interchangeable for each deletion of interest in a modular fashion (Table 1). Following the described protocol above for the generation of genetically engineered *M. marburgensis* strains, the vector construct was integrated into the genome of *M. marburgensis* site-specific to the corresponding flanking regions on the vector.

This single homologous recombination of the vector into the chromosome results in a merodiploid state where wild-type and deletion genotypes reside on the chromosome. After purification of the merodiploid mutant strain, it was transferred to non-selective MS media. Cultivating the culture under non-selective conditions enables homologous recombination between the additional copies of the upstream and downstream flanking regions on the chromosome. This resolves the merodiploid state, resulting in a mixed population of recombinants that reverted to the wild type and recombinants that exclusively keep the gene deletion state. It was shown that four generations under non-selective conditions are sufficient for homologous recombination to happen. To ensure that the recombination event takes place, four non-selective transfers (approx. 28 generations) were carried out in the first experiment. In the following experiments, it was found that one non-selective transfer (approx. seven generations) is sufficient to isolate double homologous recombined strains of *M. marburgensis*. For negative selection with 8-azahypoxanthine, non-selectively grown *M. marburgensis* were plated on solidified MS media plates. Recombinants that still maintain the *hpt* gene are sensitive to 8-azahypoxanthine, resulting in single colonies of the Δ*hpt* genotype. After successful PCR analysis, positive mutant strains were purified as described above.

## Molecular methods for analysis of *M. marburgensis* mutant strains

PCR-based analysis of *M. marburgensis* mutant strains and genomic DNA preparation was performed as stated in Fink et al. (24) with minor modifications (see Supplemental material S1 at [https://phaidra.univie.ac.at/detail/o:2120011])

## Phenotypic characterization of *M. marburgensis* Arkeon strains

We generate overnight cultures of the respective *M. marburgensis* mutant strains. From that day, cultures were inoculated in biological triplicates to $OD_{600}$ = 0.05. The triplicates were incubated at 62°C shaking at 170 rpm (Eppendorf New Brunswick Innova, Hamburg, Germany). $OD_{600}$ measurements and isoleucine quantification were performed every 3 h until 9 h of incubation and after an additional overnight incubation after 24 h in a spectral photometer (Hach Lange GmbH, Berlin, Germany).

## HPLC-MS measurement of isoleucine concentrations

In total, 25 amino acids (20 proteinogenic amino acids + homoserine, hydroxyproline, norvaline, taurine, and ornithine) were quantified directly from the growth medium. In brief, after the removal of cells, the supernatant was diluted in acetonitrile (1:3) and was warmed up for 5 min to 40°C to facilitate protein precipitation. After centrifugation (14,000 *g*; 5 min), the supernatant was mixed with the internal standard (IS) (4:1; IS was 10 $\mu mol\ L^{-1}$ of Metabolomics Amino Acid Mix purchased from Cambridge Isotope Laboratories, Inc.). For the chromatographic separation, the LC system (Agilent 1260 Infinity II Prime LC) was coupled to an MS detector (Agilent 6475 Triple Quadrupole), and separation was achieved with a Poroshell 120 HILIC-Z analytical column (2.7 µm, 2.1 × 100 mm) together with gradient elution from 90/10 to 65/35 25 mM aqu. $NH_4COO$/MeCN (pH = 3). Analytes were ionized using electrosprayionization (ESI) in positive mode and analyzed with dynamic multiple reaction monitoring (MRM). A nine-point external calibration curve with concentrations from 76 $nmol\ L^{-1}$ to 125 $\mu mol\ L^{-1}$ was used for quantification.

## Phylogenetic analysis of *cimA* genes

AnnoTree was used to identify candidate *cimA* genes with default parameters (57). The seed for the search was a KOFAM model (K09011) for (R)-citramalate synthase. Results were downloaded and searched against the *M. jannaschii* genome (DIAMOND BLAST v.2.0.15 [58]) and retained only if their best match was the characterized *cimA* from *M. jannaschii* (WP_010870909.1; NC_000909.1_1500 in GTDB [59]). This search was not intended to be exhaustive, but to provide a set of probable *cimA* sequences against

which the sequence from *M. marburgensis* could be validated. The sequences were aligned with Clustal Omega (60) (v.1.2.3) and a phylogenetic tree built using FastTree (61) (v.2.1.11) in Geneious (release 2023.2.1). Two selections of the sequences most closely related were selected for re-alignment: one most closely related to the *M. marburgensis cimA* and one most closely related to the *M. jannaschii cimA*.

A global alignment was made for the *M. jannaschii* and *M. marburgensis* sequences in Geneious (release 2023.2.1) with free end gaps using BLOSUM62 (62) as the cost matrix, a gap open penalty of 12, and a gap extension penalty of 3. A final validation was done using a reciprocal-best BLAST hit approach. The two sequences were extracted from their respective genomes and searched against the opposite using DIAMOND (v.2.0.15) to identify the top hits.

Initial tree figures (included) were generated with AnnoTree using a percent identity cut-off of 50%, but default parameters otherwise.

## ACKNOWLEDGMENTS

Research was partially funded by The COMET center ACIB: Next Generation Bioproduction, which is funded by BMK, BMAW, SFG, Standortagentur Tirol, and the Government of Lower Austria and Vienna Business Agency in the framework of COMET—Competence Centers for Excellent Technologies. The COMET-Funding Program is managed by the Austrian Research Promotion Agency FFG. Research was also partially funded by FFG through Industrienahe Dissertationen to A.S.H. (FFG NASPA [FO999901140]). Thanks to Logan Hodgskiss for proofreading the manuscript. Open access funding provided by the University of Vienna.

R.T.F., S.K.-M.R.R., and C.F. designed the experiments. M.K. and C.F. performed laboratory experimentation. A.S.H. generated the phylogenetic tree and the multisequence alignment. M.K., R.T.F., and C.F. analyzed the data. N.T. and T.S.-P. established the method for amino acid quantification and performed the analysis. A.S.H. and S.K.-M.R.R. acquired funding. S.K.-M.R.R. and C.F. supervised research. M.K. and C.F. wrote the manuscript. All authors edited the manuscript and approved the submission.

## AUTHOR AFFILIATIONS

[1]Arkeon GmbH, Tulln a.d. Donau, Austria
[2]Archaea Physiology & Biotechnology Group, Department of Functional and Evolutionary Ecology, Universität Wien, Vienna, Austria

## AUTHOR ORCIDs

Simon K.-M. R. Rittmann http://orcid.org/0000-0002-9746-3284
Christian Fink http://orcid.org/0000-0003-4233-014X

## FUNDING

| Funder | Grant(s) | Author(s) |
| --- | --- | --- |
| Austrian Centre of Industrial Biotechnology | | Simon K.-M. R. Rittmann |
| | | Christian Fink |
| Österreichische Forschungsförderungsgesellschaft | FO999901140 | Angus S. Hilts |

## ADDITIONAL FILES

The following material is available online.

## Open Peer Review

**PEER REVIEW HISTORY (review-history.pdf).** An accounting of the reviewer comments and feedback.

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
