## [Reviewer comments · Microbiology Spectrum]

Microbiology Spectrum

Markerless mutagenesis enables isoleucine biosynthesis solely from threonine in *Methanothermobacter marburgensis*

Maximilian Klein, Angus Hilts, Ross Fennessy, Nino Trattnig, Thomas Stehrer-Polasek, Simon Rittmann, and Christian Fink

Corresponding Author(s): Simon Rittmann, University of Vienna

Review Timeline:

Submission Date:	November 27, 2024
Editorial Decision:	January 31, 2025
Revision Received:	March 17, 2025
Accepted:	March 26, 2025

Editor: Amelia-Elena Rotaru

Reviewer(s): The reviewers have opted to remain anonymous.

Transaction Report:

DOI: <https://doi.org/10.1128/spectrum.03068-24>

Re: Spectrum03068-24 (Marker-less mutagenesis in *Methanothermobacter marburgensis* Arkeon verifies native pathway for Isoleucine biosynthesis via CimA and allows for integration of a synthetic pathway for Isoleucine biosynthesis from threonine through IlvA)

Dear Dr. Simon K.-M.R. Rittmann:

Thank you for the opportunity to review your work. Below are the reviewer comments - I encourage you to address their recommendations carefully.

Revision Guidelines

Sincerely,
Amelia-Elena Rotaru
Editor
Microbiology Spectrum

Reviewer #1 (Comments for the Author):

This manuscript reports a genetic method to generate markerless engineered strains of archeal methanogen *Methanothermobacter marburgensis*, a promising chassis organism for industrial biotechnology. As a proof of concept, the authors successfully applied the developed method to study and engineer isoleucine synthesis pathway in the species. Although the genetic tool developed in the manuscript (use of Hpt and 8-Azahypoxanthine) was previously established in other

methanogen species, the authors optimised this approach in *M. marburgensis*. In addition, they provided experimental verification of pyruvate-derived isoleucine synthesis in *M. marburgensis*. Furthermore, through the heterologous expression of *ilvA*, they demonstrated that *M. marburgensis* is capable of aspartate-mediated isoleucine biosynthesis. These findings are useful and important for the relevant research community.

Comments / suggestions

Line 213 - 214, 'we found mixed mutants that showed wild-type and deletion mutant signals'... showing that the selection was apparently not tight enough. However, author managed to obtain the mutant, and also implies the use of other counter-selection using 6-methylpurine for tighter selection (Line 319).

Line 215, mutant 2 instead of two? To make it consistent with designations in the Figure 3B

Regarding Figures S2 and S3, it may be beneficial to include primer binding schemes similar to those presented in Figure 3D.

Reviewer #2 (Comments for the Author):

The authors of this report have extended the use of a well-known counter selectable marker, *hpt*, to the construction of marker-less genomic changes in *Methanothermobacter marburgensis*. They have demonstrated the effectiveness of this tool in work with this organism by providing genetic analysis-based validation of the previously reported concept that a citramalate synthase-based pathway is used for isoleucine biosynthesis in many methanogens including *Methanocaldococcus jannaschii*. It is in *M. jannaschii* where a methanogen citramalate synthase gene (*cimA*) was first identified. They also demonstrated that a synthetic isoleucine biosynthesis pathway with threonine as precursor could be established in *M. marburgensis*. Thus, it is a useful report. However it has several problematic areas as listed below.

First: MANY AREAS OF THE MANUSCRIPT NEED REWRITING FOR SYNTAX ERRORS. THIS REVIEWER STOPPED at line 209 FOR PROVIDING SUGGESTIONS.

Lines 38-40: "Additionally, tools for genetic engineering of *M. marburgensis* become available that enable for sequential genetic modification." This sentence needs to be rewritten. Suggestion: "Additionally, the reported study makes the tools for genetic engineering that enable sequential genome modification in *M. marburgensis* available."

Line 52: This is to be rewritten to make sure that no reads it as if both H₂ and O₂ (not just H₂) are reacted with CO₂. It needs a qualifier, separation of H₂ and O₂.

Lines 174-178: "As the exclusive purine recycling system in methanogens, *Hpt* has been proven to be an efficient system for marker-less mutagenesis for *M. jannaschii*". To the knowledge of this reviewer *M. jannaschii* is not yet amenable to marker-less mutagenesis. The cited article "Lie, T. J. et al." (reference 25) does not concern *M. jannaschii* but *Methanocaldococcus* sp. strain FS406-22, which is distinct from the former. The cited article on *M. jannaschii* (reference 22) does not describe that *hpt* can be used as a selectable marker in this organism. Thus, the authors need to credit Lie, T. J. et al. (reference 25) for their work on *Methanocaldococcus* sp. strain FS406-22.

The authors may tone down their claims on citramalate synthase (*CimA*) as they are using the discovery of several other groups:

1999: Reference 31 represents a study on *M. jannaschii* that first time identified a methanogen *cimA*. It drew help from the following that proposed a citramalate synthase-based pathway for isoleucine biosynthesis in methanogens.

1982: Reference 32 (*Methanothermobacter marburgensis*)

1984: 10.1021/bi00303a016 (multiple methanogens)

1985: DOI: 10.1128/jb.162.3.905-908 (*Methanothermobacter marburgensis*)

Thus, the identification of the *cimA* gene in *Methanothermobacter marburgensis* is not novel in nature.

Lines 179-182: The word "integration" does not fit here well. Assimilation or incorporation would be better. In fact, one sees this case in line 182, where the word "incorporated" has been used.

Lines 188-191: "The annotated *hpt* (MTBMA_c17060) in the genome of *M. marburgensis* shares the same amino acid sequence as the characterized *hpt* gene from *M. thermotrophicus* 44. This made MTBMA_c17060 a suitable candidate gene for deletion in *M. marburgensis*." It is not what is the relevance of this sentence here.

Lines 194-195: "Additionally, we added a thermostable neomycin resistance with the weak constitutive promoter *Psynth*(BRE) and....". What is the reason for an emphasis on a "weak constitutive

promoter"?

Lines 196-200: "Although earlier studies report that deletion of hpt was achieved through transformation with constructs only harboring homologous flanking regions of hpt, we were not successful with this strategy. Thus, we added a second constitutively expressed hpt to ensure sufficient expression and therefore complete inhibition of *M. marburgensis* growth when cultivated with 8-azahypoxanthine." The rationale of this design has not been presented clearly. Is the idea being that if by chance there is a single recombination, the resultant merodiploid will be supersensitive to 8-azahypoxanthine?

A map of pArk_00004 is needed for easy comprehension of the section "Marker-less deletion of cimA results in isoleucine auxotrophy".

Similarly, for the "Complementation of CimA restores isoleucine biosynthesis pathway" section a diagram of the vector that delivered the complementation construct is needed. The description of the process or design has not been clearly presented.

Lines 207-208: "..... and a second for the vector backbone target specific at the locus of hpt on the genomic DNA of *M. marburgensis*". This part needs to be rewritten.

Lines 208-209: "With two homologous flanking regions, both up- and downstream single homologous recombination is possible." It needs a "comma" and an "a" after "and downstream". Suggestion: ".....both up- and downstream, a single homologous recombination is possible."

MANY AREAS OF THE MANUSCRIPT NEED REWRITING FOR SYNTAX ERRORS. THIS REVIEWER STOPS HERE FOR PROVIDING SUGGESTIONS.

LINES 211-214: "In this specific case, the wild-type also contains a copy of hpt, therefore no wild-type revertants were isolated. Although we did not observe wild-type revertants, we found mixed mutants that showed wild-type and deletion mutant signals in PCR (Figure 3B)." These two sentences sound contradictory. The section from here until line 220 needs a better composition.

Lines 249-250: "To prove that we do not coincidentally enrich *M. marburgensis* Δ hpt strains with intact citramalate synthase from potential subpopulations," How is this possible?

Figure S1. The quality (legibility) could be improved. There are several software available for drawing better DNA construct maps.

Figure S2: Line 193. The legend says "...outside the upstream flanking region of hpt towards inside cimA (IO), The Figure shows cimA (II) and not cimA (IO). "E. coli" needs to be italicized. These are additional examples of composition problems.

LB medium: This abbreviation needs an elaboration.

Title: Could this be shortened?

Response letter to reviewers

Dear reviewers,

the authors would like to thank you for your time and efforts to read and revise our manuscript. We are thankful for each comment that facilitates the readability and comprehension of the manuscript. We try to address every comment carefully.

Reviewer #1 (Comments for the Author):

Comments / suggestions

Line 213 - 214, 'we found mixed mutants that showed wild-type and deletion mutant signals'... showing that the selection was apparently not tight enough. However, author managed to obtain the mutant, and also implies the use of other counter-selection using 6-methylpurine for tighter selection (Line 319).

Potentially, the authors overestimated the stability of 8-azahypoxanthine since stock solutions at 4°C have been used for up to 4 weeks. In recent literature research, we found that storage of 8-azahypoxanthine is only recommended for up to 1 week at -20°C by the producer (https://www.medchemexpress.com/8-azahypoxanthine.html?srsltid=AfmBOorYhb4eIn6w6Cd4k4RzJocEOCPw6a-0zcKqzvFdQFdvI9JSTdHu&utm_source=chatgpt.com). Therefore, besides the use of 4-methylpurine, we will also elaborate on the stability of liquid stock solutions in the future.

Line 215, mutant 2 instead of two? To make it consistent with designations in the Figure 3B changed

Regarding Figures S2 and S3, it may be beneficial to include primer binding schemes similar to those presented in Figure 3D.

This is a very helpful comment, and it will clearly enhance the readability of the manuscript. Primer binding schemes were added to Figures S2 and S3.

Reviewer #2 (Comments for the Author):

First: MANY AREAS OF THE MANUSCRIPT NEED REWRITING FOR SYNTAX ERRORS. THIS REVIEWER STOPPED at line 209 FOR PROVIDING SUGGESTIONS.

The authors thank the reviewer for pointing out syntax errors. We ran a second analysis over the manuscript through Grammarly and fixed all occurred errors. Additionally, we performed a second round of proofreading by 3 native English speakers to reduce the number of syntax errors to an absolute minimum. Two of the native English speakers are co-authors. The third proof reader has been acknowledged.

Lines 38-40: "Additionally, tools for genetic engineering of *M. marburgensis* become available that enable for sequential genetic modification." This sentence needs to be rewritten.

Suggestion: "Additionally, the reported study makes the tools for genetic engineering that enable sequential genome modification in *M. marburgensis* available."

Changed

Line 52: This is to be rewritten to make sure that no reads it as if both H₂ and O₂ (not just H₂) are reacted with CO₂. It needs a qualifier, separation of H₂ and O₂.

This is a very valuable comment, and it is true that without the qualifier it leads to confusion whether oxygen is also fed to the reactor which is clearly not the case. We changed it as suggested. Thank you for the detailed review.

Lines 174-178: "As the exclusive purine recycling system in methanogens, Hpt has been proven to be an efficient system for marker-less mutagenesis for *M. jannaschii*". To the knowledge of this reviewer *M. jannaschii* is not yet amenable to marker-less mutagenesis. The cited article "Lie, T. J. et al." (reference 25) does not concern *M. jannaschii* but *Methanocaldococcus* sp. strain FS406-22, which is distinct from the former. The cited article on *M. jannaschii* (reference 22) does not describe that hpt can be used as a selectable marker in this organism. Thus, the authors need to credit Lie, T. J. et al. (reference 25) for their work on *Methanocaldococcus* sp. strain FS406-22.

Yes, Susanti et al. do not mention the negative selection while Lie et al. do and yes in that case it was *Methanocaldococcus* sp. strain FS406-22. Thank you for pointing that out! The microbe was changed from *M. jannaschii* to *Methanocaldococcus* sp. strain FS406-22 in the manuscript.

The authors may tone down their claims on citramalate synthase (CimA) as they are using the discovery of several other groups:

1999: Reference 31 represents a study on *M. jannaschii* that first time identified a methanogen cimA. It drew help from the following that proposed a citramalate synthase-based pathway for isoleucine biosynthesis in methanogens.

1982: Reference 32 (*Methanothermobacter marburgensis*)

1984: 10.1021/bi00303a016 (multiple methanogens)

1985: DOI: 10.1128/jb.162.3.905-908 (*Methanothermobacter concilii*)

Thus, the identification of the cimA gene in *Methanothermobacter marburgensis* is not novel in nature.

We do apologize if it was conceived that we want to claim the discovery of pyruvate-derived isoleucine biosynthesis in methanogens. We did not intend to claim the discovery of pyruvate-derived isoleucine biosynthesis in methanogens. We also did not intend to claim the discovery of the pyruvate-derived isoleucine biosynthesis in *M. marburgensis*. We only claim the identification of the *cimA* gene in *M. marburgensis* since it was not annotated as *cimA* in the genome before (annotated as *leuA2* in the NCBI genome of type-strain Marburg). Hence, the bioinformatic analysis of *cimA* genes across archaea. We tried to clarify the claim by differentiating between the discovery of the pathway and the identification of the *cimA* gene. CimA from *M. jannaschii* was, at least to our knowledge, only characterized heterologously

expressed in *E. coli* before. To our knowledge, it also has not been proven on a genetic level in methanogens before that deletion of *cimA* results in isoleucine auxotrophy. Due to the high level of structural similarity of LeuA and CimA, an unlikely unspecific CimA function in LeuA and vice versa could not be excluded.

To make the difference clearer between the discovery of the pathway and the identification of the gene in the manuscript, we added two more sentences and included the suggested references which are highly helpful in understanding the isoleucine biosynthesis in methanogens.

Lines 179-182: The word "integration" does not fit here well. Assimilation or incorporation would be better. In fact, one sees this case in line 182, where the word "incorporated" has been used.

changed

Lines 188-191: "The annotated hpt (MTBMA_c17060) in the genome of *M. marburgensis* shares the same amino acid sequence as the characterized hpt gene from *M. thermotrophicus* 44. This made MTBMA_c17060 a suitable candidate gene for deletion in *M. marburgensis*." It is not what is the relevance of this sentence here.

Thank you for pointing out this unclarity. We changed the sentence to highlight the relevance of the sentence more suitably.

Lines 194-195: "Additionally, we added a thermostable neomycin resistance with the weak constitutive promoter Psynth(BRE) and....". What is the reason for an emphasis on a "weak constitutive promoter"?

We tried to put as little genetic burden on the microbe as possible. Therefore, we decided to use a weak constitutive promoter for the resistance gene. We removed the word weak due to redundancy in this context.

Lines 196-200: "Although earlier studies report that deletion of hpt was achieved through transformation with constructs only harboring homologous flanking regions of hpt, we were not successful with this strategy. Thus, we added a second constitutively expressed hpt to ensure sufficient expression and therefore complete inhibition of *M. marburgensis* growth when cultivated with 8-azahypoxanthine." The rationale of this design has not been presented clearly. Is the idea being that if by chance there is a single recombination, the resultant merodiploid will be supersensitive to 8-azahypoxanthine?

In several non-successful attempts to generate the Δhpt strain with our low-efficiency transformation protocol, we always enriched spontaneous resistant 8-azahypoxanthine mutants. Due to the non-optimal selection with 8-azahypoxanthine, we decided to add a second copy of *hpt* to mimic the marker-less mutagenesis as it is performed in subsequent gene deletions anyway. We found that 8-azahypoxanthine selection works more reliably with a constitutively expressed *hpt* compared to the wild-type *hpt*. If the reason is low expression of the native *hpt* gene, regulatory effects, or anything else remains to be seen.

A map of pArk_00004 is needed for easy comprehension of the section "Marker-less deletion of cimA results in isoleucine auxotrophy".

Added as Figure S2 in the supplemental material.

Similarly, for the "Complementation of CimA restores isoleucine biosynthesis pathway" section a diagram of the vector that delivered the complementation construct is needed. The description of the process or design has not been clearly presented.

Added as Figure S3 in the supplemental material.

Lines 207-208: "..... and a second for the vector backbone target specific at the locus of hpt on the genomic DNA of *M. marburgensis*". This part needs to be rewritten.

The sentence was rewritten.

Lines 208-209: "With two homologous flanking regions, both up- and downstream single homologous recombination is possible." It needs a "comma" and an "a" after "and downstream". Suggestion: ".....both up- and downstream, a single homologous recombination is possible."

changed

**MANY AREAS OF THE MANUSCRIPT NEED REWRITING FOR SYNTAX ERRORS.
THIS REVIEWER STOPS HERE FOR PROVIDING SUGGESTIONS.**

LINES 211-214: "In this specific case, the wild-type also contains a copy of hpt, therefore no wild-type revertants were isolated. Although we did not observe wild-type revertants, we found mixed mutants that showed wild-type and deletion mutant signals in PCR (Figure 3B)." These two sentences sound contradictory. The section from here until line 220 needs a better composition.

Redundant and confusing information was removed from the text to make it better understood. With this paragraph, we wanted to address that it takes a single colony isolation and subsequent dilution series to eliminate residual wild-type signals. This could be derived from heterozygosity effects specific to *Methanothermobacter* but more likely from the cell chain formation of *Methanothermobacter marburgensis*. However, shedding light on this issue is beyond the scope of this study.

Lines 249-250: "To prove that we do not coincidentally enrich *M. marburgensis* Δ hpt strains with intact citramalate synthase from potential subpopulations," . How is this possible?

Thank you for pointing out this confusing sentence. We meant that it facilitates the PCR-based analysis after transformation when the locus of the complementation is different from the native one. Complementation at the native locus would result in the same PCR pattern and

make it impossible to distinguish between cross-contamination or else without sequencing.
The sentence was changed.

Figure S1. The quality (legibility) could be improved. There are several software available for drawing better DNA construct maps.

The font size of figure S1 was enlarged for better legibility.

Figure S2: Line 193. The legend says "...outside the upstream flanking region of hpt towards inside cimA (IO), The Figure shoes cimA (II) and not cimA (IO). "E. coli" needs to be italicized. These are additional examples of composition problems.

IO was changed to II and E. coli was italicized. Thank you for the detailed review.

LB medium: This abbreviation needs an elaboration.

Formulation has been added.

Title: Could this be shortened?

Good point. The title was shortened and made more concise.

Re: Spectrum03068-24R1 (**Markerless mutagenesis enables isoleucine biosynthesis solely from threonine in *Methanothermobacter marburgensis***)

Dear Dr. Simon K.-M.R. Rittmann:

Your manuscript has been accepted, and I am forwarding it to the ASM production staff for publication. Your paper will first be checked to make sure all elements meet the technical requirements. ASM staff will contact you if anything needs to be revised before copyediting and production can begin. Otherwise, you will be notified when your proofs are ready to be viewed.

Sincerely,
Amelia-Elena Rotaru
Editor
Microbiology Spectrum